# A Practical Approach to Assessing Physical Freshness: Utility of a Simple Perceived Physical Freshness Status Scale

**DOI:** 10.3390/ijerph19105836

**Published:** 2022-05-11

**Authors:** Okba Selmi, Danielle E. Levitt, Filipe Manuel Clemente, Hadi Nobari, Giulia My, Antonella Muscella, Katsuhiko Suzuki, Anissa Bouassida

**Affiliations:** 1Research Unit, Sportive Performance and Physical Rehabilitation, High Institute of Sports and Physical Education of Kef, University of Jendouba, Kef 7100, Tunisia; bouassida_anissa@yahoo.fr; 2High Institute of Sports and Physical Education, Ksar Said, University of Manouba, Tunis 2010, Tunisia; 3Department of Physiology, School of Medicine, Louisiana State University Health Sciences Center, New Orleans, LA 70112, USA; dlevit@lsuhsc.edu; 4Escola Superior Desporto e Lazer, Instituto Politécnico de Viana do Castelo, Rua Escola Industrial e Comercial de Nun’Álvares, 4900-347 Viana do Castelo, Portugal; filipe.clemente5@gmail.com; 5Delegação da Covilhã, Instituto de Telecomunicações, 1049-001 Lisboa, Portugal; 6Department of Physiology, School of Sport Sciences, University of Extremadura, 10003 Cáceres, Spain; hadi.nobari1@gmail.com or; 7Department of Exercise Physiology, Faculty of Educational Sciences and Psychology, University of Mohaghegh Ardabili, Ardabil 56199-11367, Iran; 8Department of Motor Performance, Faculty of Physical Education and Mountain Sports, Transilvania University of Braşov, 500068 Braşov, Romania; 9Department of Biological and Environmental Science and Technologies, University of Salento, 73100 Lecce, Italy; giulia.my@unisalento.it; 10Faculty of Sport Sciences, Waseda University, Saitama 359-1192, Japan; katsu.suzu@waseda.jp

**Keywords:** soccer, readiness, well-being, training load

## Abstract

**Background**: Monitoring physical freshness is essential in assessing athletes’ conditions during training periods, training sessions, or competitions. To date, no single physical freshness scale has been successfully validated against training load variables and widely used scales measuring different facets of physical freshness. **Objective:** In this study, we develop and test the practical utility of a perceived physical freshness (RPF) scale to monitor the condition of the athletes and to prevent excessive fatigue and insufficient recovery during training sessions or competitions. **Methods:** Sixteen professional male soccer players (mean ± SD age 26 ± 4 years) were enrolled. Training load (TL), monotony, strain, rate of perceived exertion (RPE), well-being indices (sleep, stress, fatigue, and muscle soreness), total quality recovery (TQR) and RPF were determined each day for two weeks of training, including a week intensified training (IW) and a week taper (TW). The validity of the RPF scale was assessed by measuring the level of agreement of a player’s perceived physical freshness relative to their TL variables, recovery state and well-being indices during each training phase (IW and TW) and during the overall training period (TP). **Results:** RPF increased during the TW compared to IW (ES = 2.31, *p* < 0.001, large). For the TP, IW and TW, weekly RPF was related to weekly TL (r = −0.81, r = −0.80, r = −0.69, respectively), well-being (r = −0.91, r = −0.82, r = −0.84, respectively) and TQR (r = 0.76, r = 0.91, r = 0.52, respectively), all *p* < 0.01. For the TP, IW and TW, daily RPF was related to TL (r = −0.75, r = −0.66, r = −0.70, respectively), well-being (r = −0.84, r = −0.81, r = −0.78, respectively) and TQR (r = 0.82, r = 0.81, r = 0.75, respectively), all *p* < 0.01. **Conclusions:** RPF was effective for evaluating the professional soccer players’ physical freshness and may be a strategy for coaches to monitor the physical, psycho-physiological, and psychometric state of the players before training session or matches.

## 1. Introduction

An increased training load during periods of intense training typically induces substantial neuromuscular, physiological, and hormonal responses that negatively affect well-being and recovery state [1,2]. Tapering of training after such intense training periods allows for the manifestation of improvements in physical fitness, well-being, and recovery [2]. For this reason, markers of psychometric performance associated with each training period have received much attention in recent years [3,4,5]. Questionnaires are widely used to determine soccer players’ well-being (i.e., sleep, stress, fatigue, and muscle soreness (Hooper index)), recovery state (i.e., total quality of recovery (TQR) scale) and perceived intensity (i.e., rating of perceived exertion (RPE) scale), and to monitor psycho-physiological and psychometric status. Use of these questionnaires allows for the detection of early signs of fatigue to optimize high-level training performance [1,2,3,4,5,6,7,8].

Several studies have examined well-being indices, recovery state and mood state changes associated with training periods to identify the markers that can effectively assess players’ psychometric statuses [2,6,9]. Monitoring these markers can be useful in helping coaches adjust training loads (TLs) and minimize negative outcomes resulting from intensified training periods [10]. For example, it was reported that negative psychometric status was associated with an increased training load (TL) [4,5,6,8]. It has been shown that intensified training was associated with decreased subjective measures of wellness, and mental fatigue was associated with poor recovery [5,9]. Moreover, Moalla et al. [4] showed that daily TL was related to Hooper index score in professional soccer players. Similarly, Selmi et al. [2] showed that TL during basic and intensified training periods was associated with well-being indices (i.e., sleep quality, stress, fatigue level and muscle soreness), total quality of recovery and profile of mood states (POMS) scores among professional soccer players, indicating the importance of monitoring well-being, recovery, and mood state during training periods. Ouergui et al. [1] showed that decreased TL during taper was beneficial for wellness and, importantly, physical recovery in athletes. Brink et al. [11] showed that the TQR score could predict overtraining or injury in athletes. Furthermore, Selmi et al. [12] showed that negative psychometric status and poorer perceived recovery was associated with poorer technical performance (passing, talking, ball possession and interception) in professional soccer players during the competitive period. Thus, poorer physiological and psychological recovery, or reduced physical freshness, could negatively affect technical performance during soccer-specific training, thus, leading to lower readiness and worst physical freshness.

Physical freshness, or a body’s current ability to function efficiently during physical exertion, is an important attribute to consider because soccer competition is influenced, in part, by the condition and readiness of the players [5]. According to Bafirman [13], physical freshness is the ability of the body to adjust to the physical demands placed upon it, including during training or competition, without causing excessive fatigue. Another definition of physical freshness is the ability of a person to easily perform a particular physical task with satisfactory results and without feeling too tired [13]. It is important in the development of technical and tactical skills, alongside the emotional and mental aspects of sports performance, in soccer players. Physical freshness is most assuredly a combination of physiological and psychometric parameters [2,7]. Creating a tool that simultaneously considers these important components is very attractive for coach’s prescription. Therefore, assessing athletes’ physical freshness could be a valuable tool during intensified and tapering training periods. However, in the currently available literature, no scales exist to determine physical freshness in soccer athletes. For that reason, this study aimed to create a practical approach to assess the rating of physical freshness (RPF) in soccer players and validate it against training load and psychometric indices determined during training (intensified and taper periods). We hypothesized that in soccer players, RPF would be lower during intensified training and increased during taper, and influenced by training load (TL), monotony, strain, psychometric status (sleep, stress, fatigue, delayed-onset muscle soreness (DOMS)), and TQR. The results will provide evidence for the use of assessing physical freshness using the RPF and the association between physical freshness, TL and psychometric factors.

## 2. Materials and Methods

### 2.1. Experimental Approach to the Problem

This study followed a cohort design of a 2-week period. Data collection occurred approximately 17 weeks after the 2020–2021 season began (mid-season). During the study period, the players were monitored each day for well-being, recovery, and TL using validated and reliable instruments. Physical freshness was also monitored each day. The players had been familiarized with the instruments since the beginning of the season, aiming to reduce the variability and noise.

### 2.2. Participants

Sixteen professional male soccer players (age 26 ± 4 years; height 179 ± 7 cm; body mass 74.1 ± 9.5 kg; body fat 11.7 ± 2.7%; mean ± SD) from the same national league soccer team took part in the study. Their playing positions included 2 central defenders, 3 lateral defenders, 4 defensive midfielders, 5 offensive midfielders, and 2 forwards. The following eligibility criteria were defined: (i) all the players competed for the same soccer team; (ii) all the players participated in the national championship; and (iii) free from injury or illness for two months before the study and during the study. Goalkeepers did not participate in the same training program as the other participants and, thus, were excluded from the study. The players had 15.1 ± 2.9 years of experience in competitive soccer. During the study period, the players participated in five training days and one match per week. The study was conducted according to the Declaration of Helsinki and according to the ethical standards in sport and exercise science research [14], and the protocol was approved by the research ethics committee of the High Institute of Sports and Physical Education of Kef (ISSEP-Kef), University of Jendouba, Kef, 7100, Tunisia (approval No. 011/2021).

### 2.3. Procedure

The study took place over two weeks during the mid-season (one week of intensified training and one week of tapering). The main training objective for the first week was to increase TL, and the main training objective for the second week was to decrease TL to manifest increased fitness for the next week (taper). Before beginning the experimental period, body mass and height were measured (OHAUS, Florham Park, NJ, USA) and body fat percentage was calculated according to previously published methods [15]. Throughout the 2-week training period, the duration of each training session and rating of perceived exertion (RPE) were recorded for each player 20–30 min after each training session to calculate TL. Each player completed the rating of the physical freshness (RPF) scale, the Hooper index scale, and the TQR scale before the first training session of the day to monitor physical freshness, well-being, and recovery state, respectively (Figure 1). All the players completed the scales independently and as honestly as possible. 

The training program was organized and monitored by the team coaches without influence from the experimental investigation. The program included the following 2 microcycles of 7 days each: 1 intensified training week (IW), followed by 1 tapering week (TW) (Figure 1 and Table 1). During IW, the players performed 7 training sessions across 5 days, with twice-per-day training (i.e., one session in the morning and one session in the afternoon) on 2 of the training days. On the sixth day, each player took part in a friendly match for half of the total playing time, then performed an additional 20-min training session. The players were then given one day of rest. During TW, the players performed five training sessions, one official match and one day of rest. The players who did not participate in the match were given a day of rest. The players performed a total of 12 training sessions and 2 matches across the two microcycles (Table 1).

### 2.4. Measures

#### 2.4.1. Scales to Assess Physical Freshness, Well-Being, and Recovery States

Each player was asked to complete the well-being indices (quality of sleep, fatigue, stress, and DOMS) [16], TQR scale [17] and the RPF 15 min before the first training session each day or before the match to assess the players’ well-being, recovery state and freshness state. The answers reflected the responses to the preceding training day. The answers were provided individually to avoid teammate influence.

#### 2.4.2. Rating of Physical Freshness (RPF)

Physical freshness reflects a combination of fitness and the current readiness state of players who are performing training sessions or matches; thus, it is related to psychometric status and recovery state. To create a simple and valid practical assessment of the players’ physical freshness, an RPF scale was created (Table 2). Fifteen minutes before warming up for each training session or match, each player was shown the RPF scale with verbal anchors (Figure 1) and was asked to provide a rating of his physical freshness. This method is similar to other methods that have previously been used in studies monitoring exercise training, including those using RPE [18], well-being (Hooper index; [19]), and recovery state (TQR; [17]). The RPF scale is a tool ranging from 1 to 7, where 1 indicates “very, very poor freshness” and 7 indicates “very, very good freshness”, similar to the well-being scale (i.e., sleep, stress, fatigue and DOMS) that asks the subject to rate with highly standardized verbal instructions how they perceived the physical freshness before the effort (training session or match). This form of representation was chosen because the well-being indices scale is a commonly used and easily understood measure of an athlete’s current state of pre-fatigue before training sessions or matches; thus, the perceived feeling of physical freshness should transfer well using a similar scale. To assess the players’ freshness state, the RPF scale asked a single question, which was as follows: “What is your condition now?” on a scale of 1–7, where a higher RPF score indicated a more positive state of freshness (Table 2).

Physical freshness is most relevant when players are preparing to engage in another bout of training or competition, typically coinciding with the end of the recovery period (e.g., since the previous training session or match). Therefore, fifteen minutes prior to the beginning of the warm-up for each day’s first training session or match was selected for the assessment of physical freshness. The players received standardized verbal and visual information explaining how to interpret the RPF scale and the numerical and verbal anchors contained within it. During the players’ familiarization with this scale (4 weeks before the study), each player was provided with a copy of the scale labeled “Perceived feeling of physical freshness” (Table 2), containing a simple and clear explanation of each numerical anchor. Each player individually noted his RPF. After the familiarization, the players understood the RPF scale and had no difficulty providing their RPF rating before each day’s first training session.

#### 2.4.3. Well-Being Indices (Hooper Index)

Each player responded subjectively about the quality of sleep during the preceding night, quantity of stress, fatigue level and DOMS using subjective rating scales. The scores ranged from 1 to 7, where 1 indicated “very, very low” or “good” and 7 indicated “very, very high” or “bad” [16]. The sum of these four scores (i.e., sleep, fatigue, stress, and DOMS) was summed to calculate the Hooper index (HI) [16]. A higher HI score indicates more stress, greater fatigue, more DOMS, and worse sleep quality. The well-being indices demonstrated excellent reliability, with Cronbach’s *α* ranging from 0.90 to 0.94 in the present study.

#### 2.4.4. Total Quality of Recovery (TQR)

The TQR scale was used to estimate recovery state [17], as has previously been reported [1,2]. This scale ranges from 6 to 20, where a higher score reflects better recovery. The TQR scale obtained a Cronbach’s α value of 0.91 in the present study.

#### 2.4.5. Rating of Perceived Exertion (RPE)

To assess subjective training intensity, the players reported themselves 15 to 30 min after the end of each training session or match using the Borg CR-10 scale [20]. This method has been validated for use in soccer players [21].

#### 2.4.6. Training Load Monitoring

The duration (min) of each session and corresponding RPE for each player (15–30 min after each session) were recorded to calculate the TL. Warm up, cool down and intra-session rest were included in the training session duration. The session rating of perceived exertion (s-RPE) method was used to calculate TL, monotony, and strain for each player [20,22,23]. TL for each session was calculated by multiplying session duration and RPE [20]. Daily TL was calculated by adding the total TL for each session performed on a single day. For each week, the mean daily TL divided by the standard deviation (SD) was used to calculate monotony, which reflects TL variability throughout the week [2,22]. Monotony and total weekly TL were multiplied to calculate strain, which reflects general training stress and training variability [5,23]. The mean TL, monotony and strain were calculated for IW and TW. 

#### 2.4.7. Statistical Analysis

The Statistical Package for the Social Sciences (v20.0, SPSS, SPSS Inc, Chicago, IL, USA) was used to conduct all the statistical analyses. Data are expressed as means ± SD. The assumption of normality was verified using the Kolmogorov–Smirnov test. Pearson’s product moment correlations were used to validate the RPF scale by examining the strength of the relationships between RPF, all the TL variables, well-being indices and TQR during IW, TW, and the overall study period (2-week period training). The magnitude of correlation coefficients was interpreted according to the following cutoffs: trivial (r < 0.1); low (0.1 ≤ r < 0.3); moderate (0.3 ≤ r < 0.5); large (0.5 ≤ r < 0.7); very large (0.7 ≤ r < 0.9); nearly perfect (0.9 ≤ r < 1); and perfect (r = 1) [24]. The differences between IW and TW for all the TL variables (TL, monotony, and strain), RPF, well-being indices and TQR were examined using Student’s paired *t*-tests. Cohen’s d effect sizes (ES) were used to interpret the magnitude of the differences between training weeks [23], which were as follows: trivial (0 < d ≤ 0.20), small (0.20 < d ≤ 0.50), medium (0.50 < d ≤ 0.80), or large (d > 0.80) [25]. An alpha level of 0.05 was used to determine statistical significance.

## 3. Results

Data from 192 individual sessions or matches throughout the training period (IW and TW) were included in the analyses. The training frequency and loads are shown in Table 3.

Table 4 presents the correlation of daily RPF with daily ratings of the Hooper index and its subscales, TQR, and daily TL during the 2-week training period overall, and IW and TW separately.

Across the 2-week training period (IW and TW), weekly RPF was negatively correlated with weekly TL (r = −0.81, *p* < 0.01, very large), monotony (r = −0.54, *p* < 0.01, large), strain (r = −0.51, *p* < 0.01, large), HI (r = −0.91, *p* < 0.01, very large) and positively correlated with weekly TQR (r = 0.76, *p* < 0.01, large) (Table 4). For the IW, weekly RPF was negatively correlated with weekly TL (r = −80, *p* < 0.01, large), strain (r = −0.62, *p* < 0.01, large), HI (r = −0.82, *p* < 0.01, very large) and positively correlated with weekly TQR (r= 0.91, *p* < 0.01, very large). For the TW, weekly RPF was negatively correlated with weekly TL (r = −69, *p* < 0.01, large), HI (r = −0.84, *p* < 0.01, very large) and positively correlated with weekly TQR (r= 0.52, *p* < 0.05, large).

The TL, monotony and strain were significantly greater in IW than TW (ES = 2.70, large; ES = 1.58, large; ES = 2.45, respectively, all *p* < 0.001; Figure 2). RPF and TQR were significantly lower in IW than TW (ES = 2.31, large and ES = 2.47, large, respectively, both *p* < 0.001). The HI was significantly greater (ES = 2.66, large, *p* < 0.001) in IW than TW (Figure 3).

## 4. Discussion

The present study aimed to create a practical approach to assess physical freshness (RPF) in soccer players and validate it against training load and psychometric indices determined during training (intensified and taper periods). In this study, TL, monotony, strain well-being indices (sleep, stress, fatigue level and DOMS), TQR, and RPF decreased during the TW, compared to IW. Importantly, RPF was strongly associated with TL, monotony, strain, and psychometric status (sleep, stress, fatigue DOMS and TQR) during IW and TW, indicating its validity for assessing physical freshness during these phases of training.

The reduction in TL, quantified by the s-RPE method, and the corresponding lower level of monotony and strain during TW, was concomitant with decreased RPE low loads and sufficient recovery likely reduced fatigue [16], resulting in decreased perceived exertion and an improved feeling of freshness. The results from recent investigations indicated a relationship between low monotony and strain and a low level of fatigue resulting from suitable TL [5]. Buchheit et al. (2013) [26] reported that level of fatigue and recovery state are associated with TL variables in soccer players at the professional level. Moalla et al. (2016) [4] showed that daily TLs are also related to psychometric status (sleep, stress quality, fatigue level and muscle soreness) in soccer players at the professional level. Thus, the low level of fatigue during TW likely resulted from moderate loads, sufficient recovery, and positive well-being [21]. Moreover, Selmi et al. in 2020 [2] showed that high TL, monotony, and strain during intense training periods were associated with more negative physiological and psychomotor states in soccer players.

This study demonstrated that RPF scores were lower in IW and higher in TW among professional soccer players. RPF, across the whole 2-week training period and for each training phase (IW and TW), was strongly associated with the corresponding weekly TL variables. This study supported the efficacy of using the RPF scale as simple, non-invasive, and as a useful marker for monitoring TL in soccer. Given the importance of physical freshness in the acquisition and maintenance of technical skills, this measure could be especially useful for informing and manipulating TL, according to the intent of each session. Manipulating TL according to physical freshness during specific training periods and sessions could also aid in preventing physical and mental fatigue in soccer players.

In the present study, the psychometric variables, including HI, TQR and RPF, improved from IW to TW. This result agrees with that of Ouergui et al. [1], who indicated that during taper training, fatigue was reduced, and well-being indices and recovery state were improved in judo athletes. Furthermore, it has been widely reported that psychometric status can be positively affected by lower training load variables, monotony, and strain [1,2,3,7,9,22]. The change in psychometric status during TW might reflect positive well-being, sufficient recovery and improved physical freshness at the end of the taper period, compared to the IW. Moreover, TL-induced psychological stress could influence RPF, particularly during periods of intensified training. In the present study, daily RPF was positively related to daily TQR and negatively related to daily well-being indices (sleep, stress, fatigue, DOMS and HI) in IW, TW, and across the 2-week training period. Together, these strong relationships suggest that the RPF scale could be used as a rapid assessment of overall recovery and fatigue level, and therefore markers of the current state of fitness, in soccer players during intensified or tapering training and to provide important details about the use of this tool (RPF) to adapt TLs.

While this study has described a simple physical freshness index as a new and effective approach to assess well-being and recovery state among soccer athletes, the study was not without limitations. First, the study sample size was small and included only professional male soccer players, limiting the generalizability of conclusions. Second, the relationship between physical freshness, technical performance (i.e., successful, and lost passes, tackles, and interceptions) and time-motion parameters (i.e., total distance covered, time in different intensity domains, number of sprints, etc.) should be examined. Third, while the training program was controlled for all the players, it was not possible to control any additional activities in which the players engaged outside of their training sessions. Finally, the study was conducted during the middle of the competitive period; thus, the results cannot be generalized to other training periods (e.g., preseason or post season).

To the best of our knowledge, this investigation is the first to create a new tool (RPF) to detect physical freshness during intensified training and taper periods in professional soccer players. The results of this study support the use of RPF before training and matches as a simple, non-fatiguing, non-invasive and effective measure for helping coaches, physical coaches and medical staff to assess soccer players’ perceived physical freshness statuses, which are reflective of their overall psychometric and perceived recovery statuses. Technical staff should consider that higher training loads and greater pre-training fatigue negatively affect the physical freshness of the players.

## 5. Conclusions

This study showed that RPF was strongly related to TL variables, well-being, and recovery state during intensified and tapering training periods in professional soccer players. These results offer support for the efficacy and utility of using RPF as part of an overall program to monitor athlete readiness during training periods, training sessions, or soccer matches among professional-level players. This tool may also be useful to inform training programming to prevent excessive fatigue and insufficient recovery [27].

To extend the applicability of our findings, future investigations that examine physical freshness should be conducted during different periods of the sport season and altering different aspects (i.e., physical, technical, psychological); moreover, in these studies, players of different sexes, ages and who play in different categories should be included.

## Figures and Tables

**Figure 1 ijerph-19-05836-f001:**
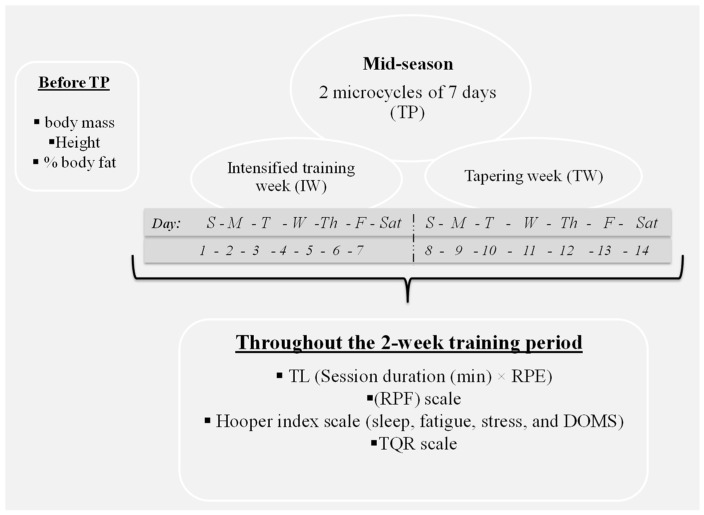
Representative diagram of the experimental protocol. S: Sunday, M: Monday, T: Tuesday, W: Wednesday, Th: Thursday, F: Friday, Sat: Saturday, TL: training load, RPE: rating of perceived exertion, RPF: rating of physical freshness, DOMS: delayed-onset muscle soreness, TQR: total quality of recovery.

**Figure 2 ijerph-19-05836-f002:**
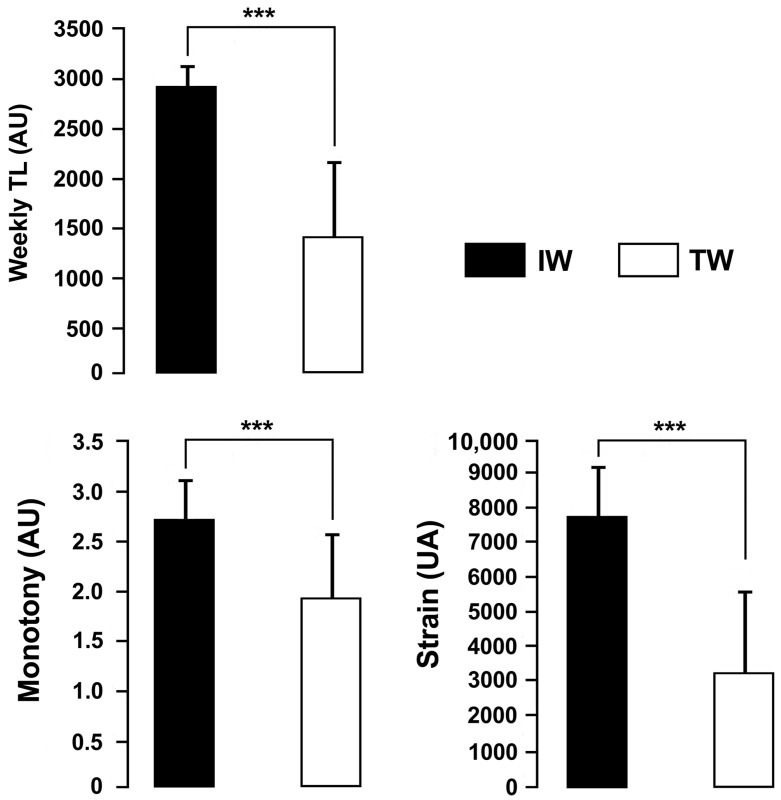
Weekly training load (TL), monotony, and strain measured throughout the intensive training week (IW) and the tapering week (TW) (mean ± SD; *n* = 16). AU: arbitrary units. *** *p* < 0.001.

**Figure 3 ijerph-19-05836-f003:**
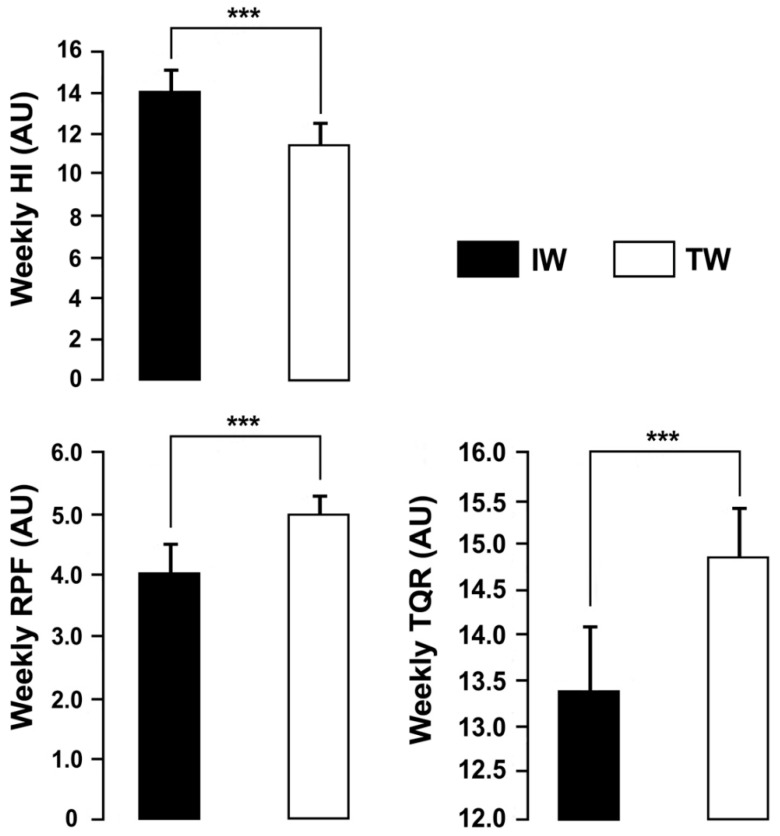
Weekly average rating of physical freshness (RPF), Hooper index (HI) and total quality of recovery (TQR), measured throughout the week of intensive training (IW) and the tapering week (TW) (mean ± SD; *n* = 16). *** *p* < 0.001.

**Table 1 ijerph-19-05836-t001:** Training time and duration, subjective intensity, training load (TL), monotony and strain across the 2-week study period (mean ± SD; *n* = 16).

		Time of Training	Duration (min)	RPE	Session-TL (AU)	Daily-TL (AU)
**Intensified week (IW)**	Sunday	15.00 h	75	3.4 ± 0.6	257.8 ± 47.2	257.8 ± 47.2
	Monday	9.00 h	70	4.4 ± 0.5	310.6 ± 35.8	565.6 ± 45.6
		16.00 h	85	3.0 ± 0.2	255.0 ± 2.2	
	Tuesday	9.00 h	75	5.4 ± 0.8	407.8 ± 6.1	739.3 ± 84.8
		16.00 h	90	3.6 ± 0.8	331.8 ± 7.4	
	Wednesday	15.00 h	95	5.8 ± 0.9	558.1 ± 84.1	558.1 ± 84.1
	Thursday	15.00 h	70	4.9 ± 0.8	345.6 ± 54.1	345.6 ± 54.1
	Friday	15.00 h	75	5.4 ± 0.7	407.8 ± 72.2	407.8 ± 72.2
	Saturday	Rest	00	00	00	00
	**Weekly-TL (AU)**	2874.7				
	**Monotony (AU)**	2.0				
	**Strain (AU)**	7885.1				
**Taper week (TW)**	Sunday	16.00 h	55	3.1 ± 0.5	168.4 ± 30.5	168.4 ± 30.5
	Monday	15.00 h	75	3.6 ± 0.6	271.9 ± 44.9	271.9 ± 44.9
	Tuesday	15.00 h	85	3.6 ± 0.6	302.8 ± 51.7	302.8 ± 51.7
	Wednesday	15.00 h	70	3.2 ± 0.5	223.1 ± 36.8	223.1 ± 36.8
	Thursdays	15.00 h	40	2.4 ± 0.6	95.0 ± 23.9	95.0 ± 23.9
	Friday	15.00 h	56	4.3 ± 2.4	307.5 ± 220.8	307.5 ± 220.8
	Saturday	Rest	00	00	00	00
	**Weekly-TL (AU)**	1337.2				
	**Monotony (AU)**	1.8				
	**Strain (AU)**	2512.4				

Abbreviations: min: minute, RPE: rating of perceived exertion, TL: training load, AU: arbitrary units.

**Table 2 ijerph-19-05836-t002:** Perceived physical freshness status.

Perceived Physical Freshness Status
1-Very, very poor freshness
2-Very poor freshness
3-Poor freshness
4-Moderate freshness
5-Good freshness
6-Very good freshness
7-Very, very good freshness

**Table 3 ijerph-19-05836-t003:** The number of training sessions, training days, and overall weekly TL during each 1-week microcycle analyzed in the present study.

	Intensified Week (IW)	Tapering Week (TW)
**Training sessions**	7	5
**Training days**	5	5
**Match days per week**	1 (friendly)	1 (official)
**Rest days**	1	1
**Average weekly TL (AU)**	>2500	<1500

Abbreviation: AU (arbitrary units).

**Table 4 ijerph-19-05836-t004:** Correlation coefficients (r) and magnitude of the correlation of daily ratings of physical freshness (RPF) with daily ratings of sleep, stress, fatigue, delayed onset muscle soreness (DOMS), Hooper index (HI), total quality of recovery (TQR) and daily training load (TL) during the 2-week training period (TP), intensified week (IW) and tapering week (TW).

			Sleep	Stress	Fatigue	DOMS	HI	TQR	TL
Daily RPF TP	r		−0.30 **	−0.25 **	−0.78 **	−0.79 **	−0.84 **	0.82 **	−0.75 **
	95%CL	Lower	−0.43	−0.39	−0.84	−0.85	−0.90	0.76	−0.81
		Upper	−0.16	−0.11	−0.71	−0.72	−0.77	0.88	−0.70
Daily RPF IW	r		−0.29 **	−0.15	−0.72 **	−0.71 **	−0.81 **	0.81 **	−0.66 **
	95%CL	Lower	−0.44	−0.34	−0.82	−0.82	−0.91	0.71	−0.76
		Upper	−0.11	−0.06	−0.62	−0.59	−0.68	0.90	−0.56
Daily RPF TW	r		−0.20 *	−0.23 *	−0.72 **	−0.76 **	−0.78 **	0.75 **	−0.70 **
	95%CL	Lower	−0.42	−0.44	−0.88	−0.84	−0.88	0.60	−0.80
		Upper	0.03	−0.02	−0.60	−0.67	−0.66	0.85	−0.58

Abbreviation: CL: confidence interval. * *p* < 0.05, ** *p* < 0.01.

## Data Availability

The data presented in this study are available on request from the corresponding author.

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
