# Peer review of "A Practical Approach to Assessing Physical Freshness: Utility of a Simple Perceived Physical Freshness Status Scale"

_ijerph, 2022, doi:10.3390/ijerph19105836_

Round 1

Reviewer 1 Report

The main purpose of the work aimed to create a practical approach to assess physical freshness (RPF) in soccer players and validate it against training load and psychometric indices determined during training.

The article is generally well written following the steps of the scientific method, based on solid literature, methods are clear, detailed and reproducible, and results are well presented and discussed in relation to the literature. The methodical part is written clearly and correctly.

The study is well designed. However I have some minor comments I’d like to express.

The Abstract must be improved, with a sequence of the following systematization: Objectives, Methods, Results, and Conclusions.

In general, my concerns about the reliability of the entire survey are raised by a small number of respondents. However, I understand the limitations of professional athlete research. Therefore, conclusions can only refer to this particular group, they should not be generalized.

Please treat the above comments as suggestions only.

Author Response

Dear editors and reviewers:

Enclosed is the revision of the manuscript ID:ijerph-1696802 "A practical approach to assessing physical freshness: Development of perceived physical freshness status scale".Thank you very much for the opportunity to revise the manuscript, as well as for the valuable and helpful comments and suggestions. We do believe that the paper has significantly improved after this revision, mainly in formatting and presentation styles as well as experimental concerns noted by the reviewers. We have modified the manuscript according to all comments and suggestions raised by the reviewers.

Reponses to reviewers

Reviewer 1

General comments

The main purpose of the work aimed to create a practical approach to assess physical freshness (RPF) in soccer players and validate it against training load and psychometric indices determined during training.

The article is generally well written following the steps of the scientific method, based on solid literature, methods are clear, detailed, and reproducible, and results are well presented and discussed in relation to the literature. The methodical part is written clearly and correctly.

Author’s response:

Thank you for the positive feedback.

Comment 1:

The study is well designed. However, I have some minor comments I’d like to express.

Author’s response:

Thank you very much for the positive feedback. The suggested revisions were addressed accordingly.

Comment 2:

The Abstract must be improved, with a sequence of the following systematization: Objectives, Methods, Results, and Conclusions.

Author’s response:

The Abstract was improved, with a sequence of the following systematization: Objectives, Methods, Results, and Conclusions.

Comment 3:

In general, my concerns about the reliability of the entire survey are raised by a small number of respondents. However, I understand the limitations of professional athlete research. Therefore, conclusions can only refer to this particular group, they should not be generalized.

Author’s response:

Thank you very much for the interesting comment. We added this as a limitation of the study.

The following sentence was added in the text: "First, the study sample size was small and included only professional male soccer players, limiting the generalizability of conclusions".

Comment 4

Please treat the above comments as suggestions only.

Author’s response:

Thank you for the feedback. We do believe that the paper has improved significantly after your valuable comments that were addressed accordingly.

Reviewer 2 Report

Reviewer’s Comments

Basically, I think that the aim and methodology of this paper are relevant to the International Journal of Environmental Research and Public Health. Unfortunately, the gap between the title and the content is too large. The reviewers have the following suggestions.

  1. According to the title and content of this study, it can be known that this is an empirical study. However, the summary only states that the use of this RPF scale is effective in assessing the physical freshness of players. The reviewer felt that the author(s) should identify the purpose of the topic and describe the process of developing the new scale in the abstract in a more rigorous manner.

  1. On page 2, about Section 1. Introduction, lines 73-88. The author(s) do a good job of explaining why a new scale was developed, making it clear to the reader its purpose. On lines 81-82, "However, to the best of our knowledge, no scales exist within the presently published literature to determine physical freshness in soccer athletes." The reviewers just want to confirm that there is currently no rating of physical freshness (RPF) scale in terms of player movement. This question is important because it affects the subsequent practice of this study.

  1. On page 4, about Section 2. Materials and Methods, on lines 149-159. The reviewers believe that the biggest problem with this study lies in this paragraph. The author's use of only a few lines of statement here suggests that it is a new scale, which is not rigorous. It is suggested that author(s) should refer to related practices in the same field, such as the development of the RPE scale. Because the development of a new scale represents that it has been verified by scientific and rigorous procedures, and can be used by others for reference in the future. It is sincerely recommended that if the topic is developing a new scale, the author should devote more attention to the development process of the scale so that the reader will not lose focus when reading this study.

After reading this study, it is found that the research topics and content vary widely. Authors are asked to confirm the topic and the real research purpose before making revisions. Developing a new scale is different from applying an already developed scale.

Author Response

Dear editors and reviewers:

Enclosed is the revision of the manuscript ID:ijerph-1696802 "A practical approach to assessing physical freshness: Development of perceived physical freshness status scale". Thank you very much for the opportunity to revise the manuscript, as well as for the valuable and helpful comments and suggestions. We do believe that the paper has significantly improved after this revision, mainly in formatting and presentation styles as well as experimental concerns noted by the reviewers. We have modified the manuscript according to all comments and suggestions raised by the reviewers.

Reviewer 2

General comments

Basically, I think that the aim and methodology of this paper are relevant to the International Journal of Environmental Research and Public Health. Unfortunately, the gap between the title and the content is too large. The reviewers have the following suggestions.

Author’s response:

Thank you for the feedback. We do believe that the paper has improved significantly after your valuable comments that were addressed accordingly.

Comment 1:

According to the title and content of this study, it can be known that this is an empirical study. However, the summary only states that the use of this RPF scale is effective in assessing the physical freshness of players. The reviewer felt that the author(s) should identify the purpose of the topic and describe the process of developing the new scale in the abstract in a more rigorous manner.

Author’s response:

Thank you for your suggestion.

Title was changed to "A Practical Approach to Assessing Physical Freshness: Utility of a Simple Perceived Physical Freshness Status Scale"

The purpose of the topic was identified and  the process of using the new scale was described  in the abstract in a more rigorous manner.

Comment 2:

On page 2, about section 1. Introduction, lines 73-88. The author or authors clearly explain why a new scale was developed, clearly explaining to the reader its purpose. At lines 81-82, "However, to our  knowledge, no scale exists in the currently published literature to determine the physical freshness of football athletes." The reviewers just want to confirm that there is currently no Physical Freshness Rating (RPF) scale in terms of player movement. This question is important because it affects the subsequent practice of this study.

Author’s response:

Thank you very much for this remark.

The reviewers confirmed that there is currently no Physical Freshness Rating (RPF) scale.

Please see the text: "However, in current literature, no scales exist to determine physical freshness in soccer athletes. ".

Comment 3:

On page 4, about section 2. Materials and methods, lines 149-159. The reviewers believe that the biggest problem with this study lies in this paragraph. The author's use of only a few lines of statement here suggests that this is a new scale, which is not rigorous. It is suggested that the authors refer to related practices in the same field, such as the development of the RPE scale. Because the development of a new scale means that it has been verified by scientific and rigorous procedures and can be used by others as a reference in the future. It is sincerely recommended that if the subject develops a new scale, the author should devote more attention to the scale development process so that the reader does not lose concentration while reading this study.

Author’s response:

Thank you very much for your comment and suggestion.

We have improved our explanation of the procedures for using RPF. Specifically, we have described and provided more detail regarding the creation of the RPF scale in this study based on other methods that were previously used in scales monitoring exercise training (e.g., RPE [Foster et al., 2001], Hooper index measuring well-being [Nobari et al., 2021]; and TQR measuring perceived recovery state [Selmi et al., 2022]). We also described how the term "Physical Freshness" was explained to the athletes.

Please see changes in the text.

Comment 4:

After reading this study, it is found that the research topics and content vary widely. Authors are asked to confirm the topic and the real research purpose before making revisions. Developing a new scale is different from applying an already developed scale.

Author’s response:

We appreciate the insightful comment. We developed the RPF scale and validated it in soccer athletes against individual scales measuring various aspects of physical freshness.  We did this during different training phases (i.e., intensified, and tapering training), providing an indication of practical utility.

Reponses to reviewers

Reviewer 3 Report

The authors compared RPF to several metrics of fatigue and training loads independently. I felt it too objective for the authors to claim and concluded that RPF may be simply used to measure these metrics combined, while some correlation coefficients (R) were not so high. 

The term "Physical Freshness" wasn't well-defined in the article. I suggest that the authors describe how “Physical Freshness” was defined in more detail. Moreover, it is important for the athletes to understand the meaning and concept of "Physical Freshness" to identify the number from the scale. The authors did not describe how the term "Physical Freshness" was explained to the athletes. 

Line 262 stated that “This study demonstrated that RPF were higher in IW and lower in TW among professional soccer players”. This is contrary to the results shown in Figure 2 and lines 230-232., which showed that RPF was lower in IW and lower in TW. 

Author Response

Dear editors and reviewers:

Enclosed is the revision of the manuscript ID:ijerph-1696802 "A practical approach to assessing physical freshness: Development of perceived physical freshness status scale". Thank you very much for the opportunity to revise the manuscript, as well as for the valuable and helpful comments and suggestions. We do believe that the paper has significantly improved after this revision, mainly in formatting and presentation styles as well as experimental concerns noted by the reviewers. We have modified the manuscript according to all comments and suggestions raised by the reviewers.

Reponses to reviewers

Reviewer 3

 Comment 1:

The authors compared RPF to several metrics of fatigue and training loads independently. I felt it too objective for the authors to claim and concluded that RPF may be simply used to measure the semetrics combined, while some correlation coefficients (R) were not so high..

Author’s response:

Thank you for the positive feedback.

Concerning the correlation: Daily RPF was correlated with DOMS, fatigue, HI and TQR with a magnitude of correlation coefficients is very large but with sleep and stress, the magnitude of correlation coefficients is low. Which explains that  RPF responds more to muscle pain, fatigue, and recovery than stress and sleep.

Weekly RPF was correlated with the variables of training load and well-being indices and TQR with a magnitude of correlation coefficients varies between large and very large. Which explains that the bad RPF is very related to the increase in fatigue, the weak recovery is the increase in the training load.

Comment 1:

The term "Physical Freshness" wasn't well-defined in the article. I suggest that the authors describe how “Physical Freshness” was defined in more detail. Moreover, it is important for the athletes to understand the meaning and concept of "Physical Freshness" to identify the number from the scale. The authors did not describe how the term "Physical Freshness" was explained to the athletes.

Author’s response:

Thank you very much for the interesting suggestion.

The term “Physical Freshness” was defined in more detail in the text (introduction).

More details were added to describe how the term "Physical Freshness" was explained to the athletes.(Methods: Rating of physical freshness (RPF)

Comment 1:

Line 262 stated that “This study demonstrated that RPF were higher in IW and lower in TW among professional soccer players”. This is contrary to the results shown in Figure 2 and lines 230-232., which showed that RPF was lower in IW and lower in TW.

Author’s response:

Thank you very much for the remark.

We apologize for the typo; we have corrected this error.

Please find changes in the text :

Results part : "RPF and TQR were significantly lower in IW than TW (ES=2.31, large and ES=2.47, large, respectively, both  p<0.001)" and discussion part: paragraph 3 "This study demonstrated that RPF were lower in IW and higher in TW among  professional soccer players"

Round 2

Reviewer 2 Report

Reviewer’s Comments

  1. The description of the abstract should be improved. For example, On page 1, lines 38-40. " For the TP, IW and TW, weekly 38 RPF was related to weekly TL variables, well-being and TQR, all p<0.01. For the TP, IW and TW, 39 daily RPF was related to TL, well-being and TQR, all p<0.01. " The reviewers suggest that authors should show both significant correlation coefficients and p-values.

  1. In order to provide readers with a clearer understanding of the research, it is suggested that the author(s) may add a research structure diagram or research flow on pages 3 or 4.

  1. On page 10, about Section 5. Conclusions, lines 341-346. It is suggested that the author(s) can add future research directions or further research and verification issues.

Author Response

=============================================================

REVIEW

=============================================================

Manuscript ID: ijerph-1696802

Dear editors and reviewers:

Enclosed is the revision of the manuscript ID: ijerph-1696802 " A Practical Approach to Assessing Physical Freshness: Utility of a Simple Perceived Physical Freshness Status Scale". Thank you very much for the opportunity to revise the manuscript, as well as for the valuable and helpful comments and suggestions. We do believe that the paper has significantly improved after this revision, mainly in formatting and presentation styles as well as experimental concerns noted by the reviewers. We have modified the manuscript according to all comments and suggestions raised by the reviewers.

Reponses to reviewer

Reviewer 2

Comment 1:

The description of the abstract should be improved. For example, On page 1, lines 38-40. " For the TP, IW and TW, weekly 38 RPF was related to weekly TL variables, well-being and TQR, all p<0.01. For the TP, IW and TW, 39 daily RPF was related to TL, well-being and TQR, all p<0.01. " The reviewers suggest that authors should show both significant correlation coefficients and p-values.

Author’s response:

Thank you very much for the suggestion. significant correlation coefficients and p-values were added in the abstract

Comment 2:

In order to provide readers with a clearer understanding of the research, it is suggested that the author(s) may add a research structure diagram or research flow on pages 3 or 4.

Author’s response:

Thank you very much for the suggestion. Research structure diagram was added on page 3 (Figure 1)

Comment 3:

On page 10, about Section 5. Conclusions, lines 341-346. It is suggested  that the author(s) can add future research directions or further research and verification issues.

Author’s response:

Thank you very much for the interesting suggestion. We added add future research directions The following paragraph was added in the text:

"To extend the applicability of our findings, future investigations examining physical freshness should be conducted during different periods of the sport season and altering different aspects (i.e., physical, technical, psychological); moreover, in these studies players of different sexes, ages and who play in different categories should be included."